# Kinesiophobia and its correlations with pain, proprioception, and functional performance among individuals with chronic neck pain

Faisal Asiri[1], Ravi Shankar Reddy[1]*, Jaya Shanker Tedla[1], Mohammad A. ALMohiza[2], Mastour Saeed Alshahrani[1], Shashikumar Channmgere Govindappa[3], Devika Rani Sangadala[1]

1 Department of Medical Rehabilitation Sciences, College of Applied Medical Sciences, King Khalid University, Abha, Saudi Arabia, 2 Department of Rehabilitation Sciences, College of Applied Medical Sciences, King Saud University, Riyadh, Saudi Arabia, 3 Department of Physical Therapy, College of Applied Medical Sciences, University of Hail, Hail, Saudi Arabia

* rshankar@kku.edu.sa

**Data Availability Statement:** All relevant data are within the paper and its Supporting Information files.

## Abstract

Chronic neck pain (CNP) incidence in the general population is high and contributes to a significant health problem. Kinesiophobia (fear of pain to movement or re-injury) combined with emotions and physical variables may play a vital role in assessing and managing individuals with CNP. The study's objectives are 1) to evaluate the relationship between kinesiophobia, neck pain intensity, proprioception, and functional performance; 2) to determine if kinesiophobia predicts pain intensity, proprioception, and functional performance among CNP individuals. Sixty-four participants with CNP (mean age 54.31 ± 9.41) were recruited for this cross-sectional study. The following outcome measures were evaluated: Kinesiophobia using the Tampa Scale of Kinesiophobia (TSK), neck pain intensity using the visual analog scale (VAS), cervical proprioceptive joint position errors (in flexion, extension, and rotation directions) using cervical range of motion (CROM) device and handgrip strength as a measure of functional performance using the Baseline® hydraulic hand dynamometer. Kinesiophobia showed a strong positive correlation with neck pain intensity ($r = 0.81$, $p<0.001$), a mild to a moderate positive correlation with proprioception joint position errors (JPE) in extension, rotation left and right directions ($p<0.05$), but no correlation in flexion direction ($p = 0.127$). Also, there was a moderate negative correlation with handgrip strength ($r = -0.65$, $p<0.001$). Regression analysis proved that kinesiophobia was a significant predictor of pain intensity, proprioception, and functional performance ($p<0.05$). This study infers that kinesiophobia in individuals with CNP predicts pain, proprioception, and functional performance. Kinesiophobia assessment should be considered in regular clinical practice to understand the barriers that can influence rehabilitation outcomes in CNP individuals.

**Funding:** The King Khalid University, Abha, Kingdom of Saudi Arabia (Grant number: RGP.1/98/42) funded this study. Funding was received by the first author, Dr. Faisal Asiri.

**Competing interests:** No authors have competing interests.

## Introduction

In the general population, chronic neck pain (CNP) is one of the debilitating conditions that can impair the ability to perform regular everyday activities, decrease productivity, and adversely affect life quality [1]. In developed countries, approximately two-thirds of people experience neck pain [1]. At a given point in time, about 14% to 16% of the adult population globally experience neck pain [2], and the mean lifetime prevalence is 48.5% [2].

Chronic pain is categorized as pain that lasts more than three months [3]. However, there are distinct pathological mechanisms that contribute to the development of chronic musculo-skeletal pain. It is critical to understand neuroplasticity (a neuron's capacity to completely alter its structure, function, or biochemical profile in response to repeated afferent sensory inputs) to know how acute pain transforms as chronic pain [4]. Local inflammation of the injured tissue increases peripheral sensory neurons' sensitivity (nociceptors), resulting in repetitive abnormal afferent input to the central nervous system [5]. Researchers also discovered that people with chronic pain have less volume in their prefrontal cortex—the part of the brain that controls thoughts, personality expression, and social behavior [6]. Chronic pain has been shown to induce escape and avoidance behaviors and is strongly associated with kinesiophobia [6, 7].

Kinesiophobia is a concept that describes a condition in which a patient has an unwarranted and deteriorating fear of physical movement and actions that results from a feeling of vulnerability to painful injury or re-injury [7]. An exaggerated negative cognitive and affective response to an anticipated or actual pain is expressed as pain catastrophizing [8]. It is characterized by an increase in the possible negative aspects of pain, an inability to disengage from stressful thinking, and a sense of helplessness in dealing with pain. In the acute pain stage, these habits may be adaptive [8]. However, in long-lasting pain, the issue paradoxically worsens, aggravating impairment and pain perception thresholds as patients enter a vicious cycle that perpetuates chronic pain and functional disability [9, 10].

CNP is multifactorial, and the factors that contribute to maintain and increase pain intensity are hard to define [11, 12]. A closed-loop, proprioceptive, vestibular, and visual systems' interplay role works to maintain static and dynamic balance during functional tasks [11, 13]. Kinesiophobia and catastrophic behaviors can induce neck pain recurrence or cause changes in the somatosensory system [14]. The cervical afferent input to the higher centers may be changed by these changes, thereby impairing the cervical proprioception, which warrants assessment in detail [11]. Previous studies have shown that CNP individuals may be difficult or incapable of performing functional tasks [15, 16]. Also, kinesiophobia can further hinder their overall functional performance, which can affect their quality of life [17, 18]. Due to the avoidance of physical exercise, kinesiophobia may contribute to a deterioration of functional ability, leading to decreased mobility and chronic pain [18]. However, there is no conclusive evidence, how kinesiophobia impacts functional performance among individuals with CNP.

It is vital to develop effective recovery strategies in these neck pain patients by evaluating psychological factors before and after rehabilitation and recognizing which psychological impairments contribute significantly to neck rehabilitation [19]. There is no explored research, to our knowledge, in neck pain patients that correlated pain, proprioception and functional performance with kinesiophobia. The objectives of this study are 1) to evaluate the relationship between kinesiophobia, neck pain intensity, proprioception, and functional performance; 2) to determine whether kinesiophobia predicts pain intensity, proprioception, and functional performance among CNP individuals. We hypothesize that kinesiophobia is significantly associated with pain, proprioception, and functional performance. This study's results may provide a fundamental understanding of the interactions between kinesiophobia, pain, proprioception, functional performance, and clinical management characteristics of kinesiophobia in CNP patients.

## Material and methods

### Design and settings

This cross-sectional study was conducted in the Department of Medical Rehabilitation, King Khalid University. The study followed the 1975 Declaration of Helsinki guidelines, as amended in 1983. King Khalid University ethics committee board (ECM#2019–61) approved this study. All the individuals read the patient information sheet, which included a concise summary of the research objectives and a detailed description of the research process and signed written consent before the study's commencement. The individual pictured in Fig 1 has provided written informed consent to publish their image alongside the manuscript.

### Subjects

Sixty-four participants (38 males and 26 females) included in this study were over 18 years (age range: 28–64 years) and were referred to the physical therapy department by an orthopedist or general physician. Neck pain was defined as discomfort felt dorsally between the occiput's (inferior margin) and T1. If they met the following inclusion criteria, CNP individuals were recruited: 1) neck pain for more than three months; 2) Chronic neck pain elicited by neck postures, neck movements, or palpation of the cervical musculature; 3) neck pain intensity of 30 to 70 mm measured on a visual analog scale (VAS); and 4) neck disability score, 15 or more measured on neck disability index (NDI). Exclusion criteria included: 1) signs of radiculopathy that were tested and confirmed by a positive Upper Limb Tissue Stress Test and Spurling Test [20]; 2) history of neurological disease or whiplash injury; 3) cervical myelopathy; 4) tumors; 5) cervical spine infection, and 6) insufficiency of the vertebrobasilar artery.

### Sample size calculation

We used G*power 3.1 software (Universities, Dusseldorf, Germany) [21] to compute the study sample. As this study's primary outcome, the Tempa scale of kinesiophobia (TSK) score (mean and standard deviation) was used to estimate the study sample [9]. The calculated sample size was 64, using a power of 0.80, an alpha of 0.05, and a beta of 0.2.

### Outcome measures

**Anthropometric data.**   Following the standard protocol, anthropometric characteristics were measured, including height (m), weight (kg), and BMI (kg/m2) (Table 1).

### Kinesiophobia assessment

The Tampa Scale of Kinesiophobia (TSK) assessed the fear of movement or re-injury [22]. There was an appropriate degree of internal consistency in this questionnaire's original versions (Cronbach's alpha of 0.8), proof of prejudice, and parallel criterion-related and incremental validity [22]. The total TSK score ranges from 17 to 68, where 17 indicates no kinesiophobia, 68 shows moderate kinesiophobia, and ± 37 indicates that kinesiophobia is present [22].

**Pain assessment.**   The current neck pain intensity was assessed using the visual analog scale (VAS) scores. The scale is 100 mm long and is anchored by the words "no pain" and "worst pain imaginable "on the left and right sides [23]. Individuals were requested to draw a vertical mark that better reflects the pain level across the horizontal line; 0–30 mm indicates mild, 30–70 mm indicates moderate, and > 70 mm indicates severe pain intensity. VAS is a widely used evaluation method and has good reliability and validity [24].

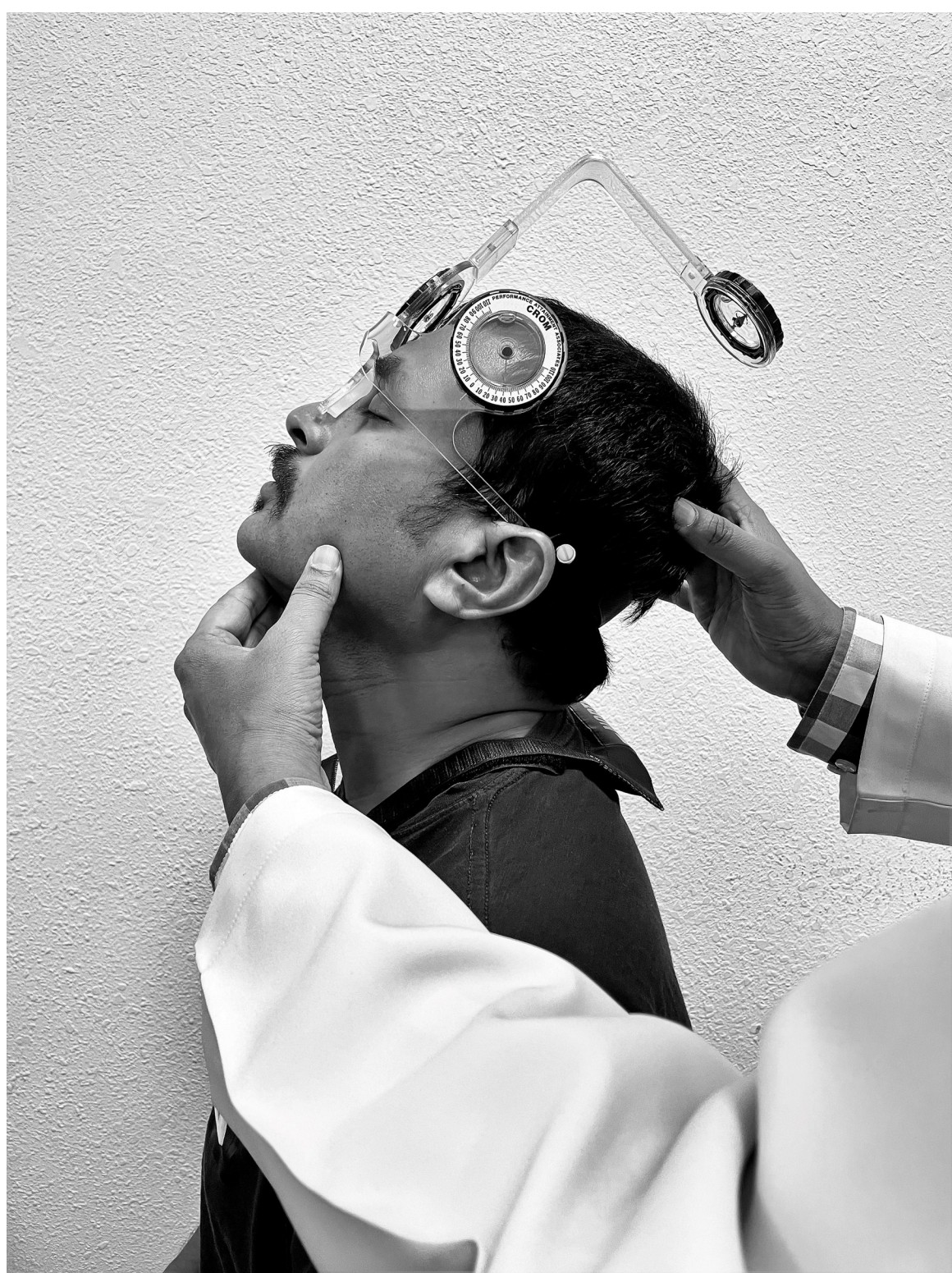

**Fig 1. Cervical proprioception assessment using a cervical range of motion device.**

**Table 1. Participant demographics, self-report measures, and study data (*n* = 64).**

| Variables | Mean ± SD (range) |
|---|---|
| Age (years) | 54.31 ± 9.41 (28–64) |
| Gender (Male: Female) | 38: 26 |
| BMI (kg/m2) | 25.60 ± 2.34 (21.4–26.1) |
| VAS (mm) | 56.2 ± 0.98 (2.2–7.9) |
| NDI | 19.65 ± 3.67 |
| TSK total score | 49.73 ± 11.82 (21–65) |
| Handgrip strength (kg) | 21.06 ± 8.10 (5–38) |
| JPE | |
| • Flexion | 1.49 (2–8) |
| • Extension | 1.14 (4–9) |
| • Rotation Left | 1.391 (3–8) |
| • Rotation Right | 5.16 ± 1.336 (3–8) |

BMI = body mass index; VAS = visual analog scale; NDI = Neck Disability Index; TSK = Tampa scale of kinesiophobia; JPE = joint position error.

**Proprioception testing.** The Cervical Range of Motion (CROM) device (Performance Attainment Associates, Minnesota) was used to assess cervical spine range of motion (ROM) and proprioception. The CROM unit has three inclinometers and a magnetic yoke or harness (Fig 1). The individual in this manuscript has given written informed consent to publish these case details. The principles for joint position errors (JPE) measurements have been adopted by Alahmari et al. [25]. The JPE is estimated according to the participant's capacity to consciously reposition his or her head to a target location previously shown by the examiner. Each individual was guided to the testing lab and got familiarized with the testing procedures. The individuals were asked to sit in the chair and put on the CROM device as if it were a pair of glasses, which was then secured around the head using the Velcro band (Fig 1). The individual in this manuscript has given written informed consent to publish these case details. A magnetic yoke was positioned directly over the participant's shoulders and pointed north. The examiner used a webbing strap to minimize the patients' shoulder and trunk motions during the examination. The examiner asked the individual to maintain the head in the neutral position (starting point) and standardized the CROM device to the starting position.

To start with JPE testing, the participants were asked to close their eyes throughout the testing procedure. The examiner slowly guided the participant's head to the target position, which was previously determined and is 50% of their maximum ROM [13]. The participant's head was then held in the target position for three seconds, allowing the individuals to memorize the target position. Successively, the examiner brought the participant's head back to a starting position. The participant was then asked to reposition their heads to the target position consciously (absolute error). The examiner measured the relocation accuracy in degrees once they reached the reference position. The JPE testing was performed in four directions, i.e., flexion, extension, left rotation, and right rotation. A simple chit method was used to randomize the order of JPE testing in four directions [26]. Three attempts were performed in each movement direction, and an average of three attempts was used for analysis.

**Functional performance.** Handgrip strength as a measure of functional performance was measured using the Baseline® hydraulic hand dynamometer. It is a valid, clinically easy, useful test to measure grip strength [27] and identify any functional performance changes in individuals with CNP. This test is performed with the participant sitting in the chair with the

shoulder adducted and neutrally rotated, elbow flexed to 90 degrees, forearm and wrist maintained in the neutral position (neither flexed nor extended), and gripping the handheld dynamometer (Fig 2) [28]. The individual in this manuscript has given written informed consent to publish these case details. The dynamometer was set to the second or third handle position to ensure consistency, claimed to be more suitable by the participant [29]. For most participants, the second position was used, which was considered the optimal level for grip evaluation and was adopted for routine testing by the American Society of Hand Therapists [29]. The individuals squeezed the handheld dynamometer's handle as hard as possible, as explained by the investigator. The measurements were performed on the dominant side, three trials were conducted, and an average of three trials was used for analysis. The handgrip strength was recorded in kilograms. Between each attempt, a one-minute rest period was allowed to minimize fatigue effects. No verbal encouragement was to the participants during the handgrip strength measurements. The hydraulic hand dynamometer was calibrated regularly throughout the study duration.

## Statistical analysis

Descriptive demographic statistics, correlation, and regression analysis were performed using the IBM SPSS statistical software version 20 (IBM Corporation, USA). The significance of the study critical values was set at $p < 0.05$. Using the Shapiro-Wilks test, the normality of the study variables was analyzed. Pearson's correlation coefficient was used to assess the relationship between kinesiophobia and pain, proprioception, and functional performance, and correlations were interpreted as follows: $< 0.3$ = mild, 0.31–6.9 = moderate, and $\geq 0.7$ = strong. Furthermore, multivariate linear regression analysis was performed to determine whether kinesiophobia predicts pain, proprioception, and functional performance.

## Results

Table 1 summarizes the demographic characteristics (age, BMI), pain intensity, proprioception JPE's and physical performance (handgrip strength) values of the study population. Sixty-four CNP individuals participated in the study. The TSK score of this study's participants was 49.73.

The correlations between kinesiophobia and neck pain intensity, proprioception, and functional performance are summarized in Table 2 and Fig 3. To our expectation, kinesiophobia showed a strong positive correlation with neck pain intensity (r = 0.81, p<0.001), a mild to

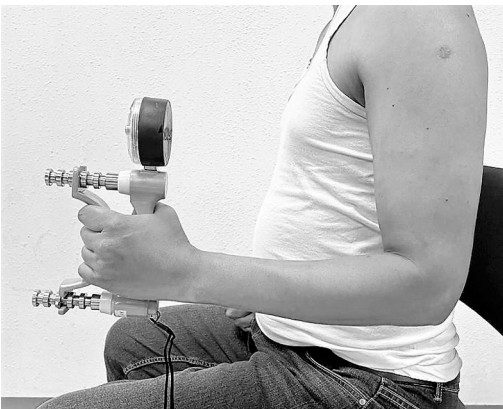

**Fig 2. Handgrip strength evaluation using a handheld dynamometer.**

**Table 2. Relationship between kinesiophobia, pain intensity, functional performance, and proprioception (*n* = 64).**

| Correlated Variables | Kinesiophobia | |
|---|---|---|
| | *r* | *p* value |
| Pain intensity (VAS) | 0.81 | <0.001 |
| JPE | | |
| • Flexion | 0.19 | 0.127 |
| • Extension | 0.48 | <0.001 |
| • Rotation Left | 0.28 | 0.025 |
| • Rotation Right | 0.31 | 0.011 |
| Handgrip strength (kg) | -0.65 | <0.001 |

VAS = visual analog scale; JPE = joint position error. The correlation was tested using Pearson's correlation coefficient analysis.

moderate positive correlation with proprioception JPE errors in extension, rotation left and right directions ($p<0.05$), but no correlation in flexion direction ($p = 0.127$). Also, there was a moderate negative correlation with handgrip strength ($r = -0.65$, $p<0.001$).

Table 3 shows multivariate linear regression analysis findings between kinesiophobia, neck pain intensity, proprioception JPE and functional performance. Kinesiophobia was a significant predictor of neck pain intensity ($r^2 = 0.53$, $p<0.001$), proprioception (extension: $r^2 = 0.12$, $p<0.002$; rotation right: $r^2 = 0.09$, $p<0.008$) and functional performance ($r^2 = 0.41$, $p<0.001$) in individuals with CNP.

## Discussion

This study intended to establish the relationship between kinesiophobia, neck pain intensity, functional performance, and proprioception in CNP individuals. This study adds a crucial dimension to the findings of research on CNP individuals. The kinesiophobia showed a significant positive correlation with pain intensity and proprioceptive JPE's and a significant negative correlation with handgrip strength. This study also showed that kinesiophobia significantly predicted neck pain intensity, cervical proprioception, and functional performance in individuals with CNP.

To the best of our knowledge, research assessing the association of kinesiophobia with pain intensity, cervical proprioception, and functional performance in populations with CNP is limited in the scientific literature. A previous study by Gunay et al. [30] showed no correlation between kinesiophobia and pain intensity in 87 non-specific CNP individuals. However, our study demonstrated a significant correlation between kinesiophobia and pain intensity. Compared to Gunay et al.'s study, our study participants had increased mean pain intensity (56.2 on VAS) and TSK score (59.73), which might have influenced these results. This study participants' mean age was higher (54 years), and elderly individuals with CNP may show less resilience towards pain intensity and kinesiophobia [9]. Thus, this might explain the significant correlation between kinesiophobia and pain intensity in the study. The literature on kinesiophobia in other populations with chronic pain is more robust. Like this study's results, Comachio et al. [31], using kinesiophobia in chronic back pain individuals, found a strong association between TSK scores and pain severity, disability, and quality of life [31]. Vaegter et al. [32], in an explorative analysis, showed a higher degree of kinesiophobia with increased pain intensity, and vice versa, in chronic musculoskeletal pain individuals [32].

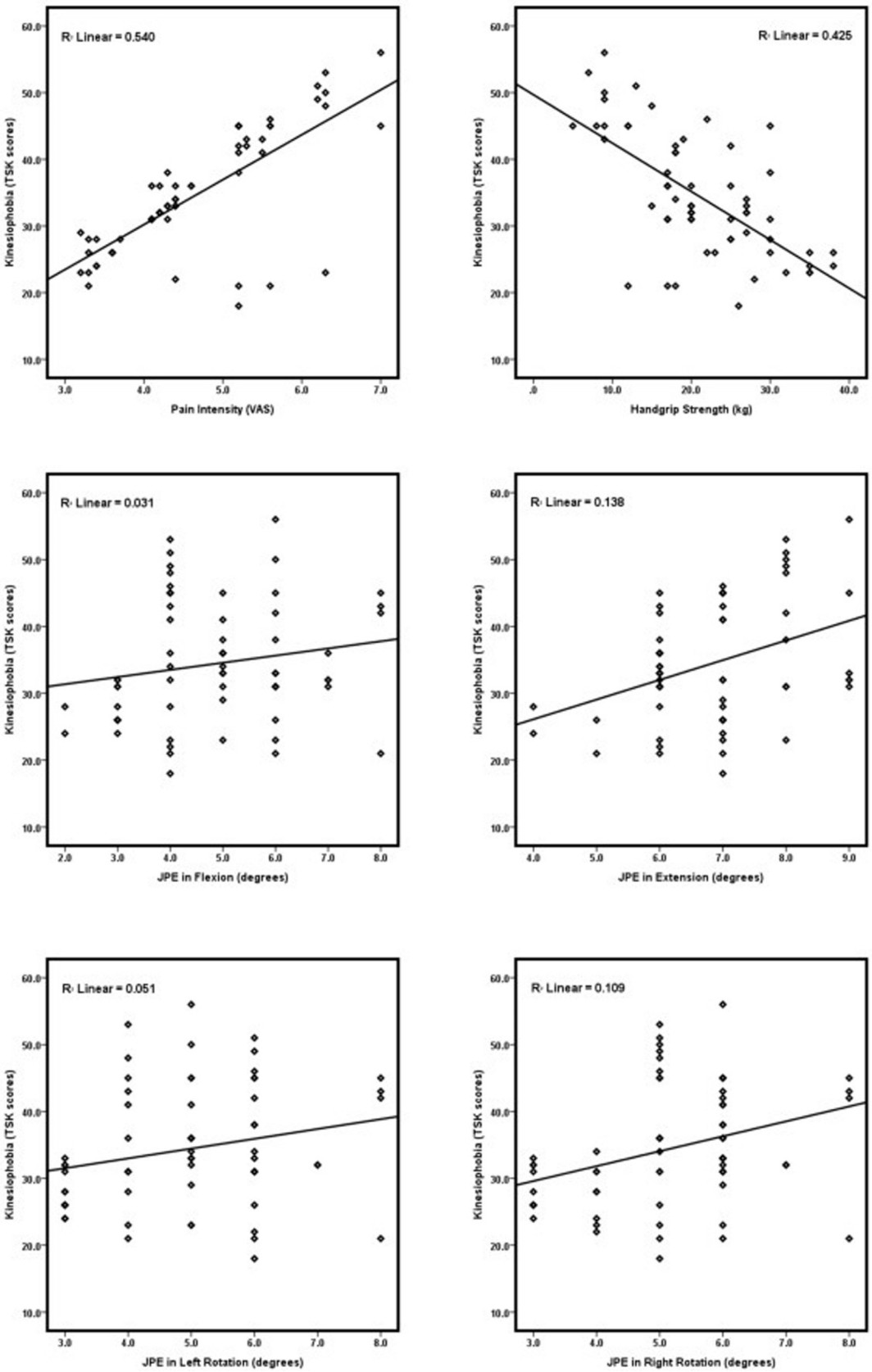

**Fig 3. The relationship between kinesiophobia, pain intensity, functional performance, and proprioception JPEs in flexion, extension, left and right rotation.** (*n* = 64). VAS = Visual analog scale; TSK = Tempa scale of kinesiophobia; JPE = Joint position error.

This study showed a mild to moderate association between TSK scores and JPE's in extension and rotation directions, fitting with our hypothesis, indicating that proprioceptive behavior is associated with fear of movement. The most direct inference on this finding may be that the cervical muscles' proprioceptive functioning, considered the most significant afferent source of the neck, performing independently of its structure, and force-generating capabilities are influenced by fear of movement [11, 33]. There is a vicious cycle of maladaptive thoughts, leading to immobility with an increased perception of pain, leading to muscle atrophy and fibrosis, thus translating to functional disability [14]. During movements, kinesiophobia can also influence the altered activation patterns of neck muscles that may modify afferent input, leading to impaired proprioception [34]. In individuals with chronic low back pain, Pakzad et al. reported that pain catastrophizing was correlated with altered motor control and activation patterns [35]. Similar relationships may exist in CNP individuals.

Kinesiophobia induced by pain stimulation ultimately leads to fear avoidance behavior that can impact functional performance, including upper limb functions [9]. This study found a significant association between kinesiophobia and handgrip strength among individuals with CNP. A previous study [9] revealed that the crucial predictors of upper extremity function and disability significantly correlated with handgrip strength and kinesiophobia [9]. Bartlett et al. [36] showed a significant correlation between TSK scores, catastrophizing, disability, and comorbidity of musculoskeletal complaints in their investigation on individuals with neck and shoulder disorders [36]. The positive correlation in this study may be due to fear of movement. Long-term inhibition of upper extremity muscles may lead to disuse atrophy and translate to decreased handgrip strength [34].

## Limitations

This study design was cross-sectional; therefore, causal relationships proving the presence of a temporal sequence between the exposure factor and the subsequent development of the disease cannot be identified. Cross-sectional research could, however, be performed as the first phase of a cohort study. This study limits the generalizability of our results, as this study included patients with moderate neck pain and disability. In this study, only absolute errors were recorded; constant and variable errors were not measured. The constant error and variable errors would have given more meaningful information regarding the direction and magnitude

**Table 3. Multivariate linear regression of TSK and explanatory variables (*n* = 64).**

| Variable | Adjusted R square | B | SE | *p* value |
|---|---|---|---|---|
| Pain intensity | 0.53 | 0.73 | 0.01 | <0.001 |
| JPE | | | | |
| • Flexion | 0.02 | 0.18 | 0.21 | |
| • Extension | 0.12 | 0.37 | 0.15 | |
| • Rotation Left | 0.04 | 0.23 | 0.20 | |
| • Rotation Right | 0.09 | 0.33 | 1.18 | 0.008 |
| Handgrip strength (kg) | 0.41 | -0.65 | 0.08 | <0.001 |

JPE = joint position error; B = Standardized Coefficients Beta, SE = standard error.

of errors in proprioception tests. Furthermore, confounding variables such as age, gender, nature of the job, marital status, formal education, smoking, sleeping hours, and leisure time (sport and hobby) that may have affected these findings were not considered by the authors. As a result, future studies should look at these confounding variables and see how they affect the outcome.

## Conclusion

In conclusion, this study demonstrated that kinesiophobia significantly correlated with pain intensity, proprioception, and functional performance in CNP individuals. Also, kinesiophobia predicted pain, proprioception, and functional performance. Therefore, these correlations should be performed in clinical settings to understand and manage CNP individuals and further validate the current findings.

## Supporting information

**S1 Data. Study protocol.**
(DOCX)

**S2 Data.**
(XLSX)

**S1 File.**
(PDF)

## Acknowledgments

We thank the deanship of scientific, King Khalid University research for their support in presenting this research.

## Author Contributions

**Conceptualization:** Faisal Asiri, Ravi Shankar Reddy.

**Data curation:** Jaya Shanker Tedla, Devika Rani Sangadala.

**Formal analysis:** Ravi Shankar Reddy, Mohammad A. ALMohiza, Mastour Saeed Alshahrani.

**Funding acquisition:** Faisal Asiri.

**Investigation:** Faisal Asiri, Ravi Shankar Reddy, Shashikumar Channmgere Govindappa, Devika Rani Sangadala.

**Methodology:** Faisal Asiri, Ravi Shankar Reddy, Mohammad A. ALMohiza, Mastour Saeed Alshahrani.

**Project administration:** Faisal Asiri, Ravi Shankar Reddy.

**Resources:** Mohammad A. ALMohiza, Mastour Saeed Alshahrani, Shashikumar Channmgere Govindappa, Devika Rani Sangadala.

**Supervision:** Ravi Shankar Reddy, Jaya Shanker Tedla.

**Writing – original draft:** Ravi Shankar Reddy, Shashikumar Channmgere Govindappa.

**Writing – review & editing:** Faisal Asiri, Ravi Shankar Reddy, Jaya Shanker Tedla, Mohammad A. ALMohiza, Mastour Saeed Alshahrani, Shashikumar Channmgere Govindappa, Devika Rani Sangadala.

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
