## [Decision Letter · Decision Letter 0]

30 Mar 2021

PONE-D-21-05740

Kinesiophobia and its correlations with pain, proprioception, and functional performance among subjects with chronic neck pain

PLOS ONE

Dear Dr. REDDY,

Thank you for submitting your manuscript to PLOS ONE. After careful consideration, we feel that it has merit but does not fully meet PLOS ONE’s publication criteria as it currently stands. Therefore, we invite you to submit a revised version of the manuscript that addresses the points raised during the review process.

We look forward to receiving your revised manuscript.

Kind regards,

Bernadette Ann Murphy, PhD

Academic Editor

PLOS ONE

Additional Editor Comments:

Reviewer 2 has identified some major concerns. The authors are encouraged to carefully address the concerns of both reviewers if they wish to consider resubmitting.

Journal Requirements:

4. We note that Figure 1 and 2 include an image of a patient. 

As per the PLOS ONE policy (http://journals.plos.org/plosone/s/submission-guidelines#loc-human-subjects-research) on papers that include identifying, or potentially identifying, information, the individual(s) or parent(s)/guardian(s) must be informed of the terms of the PLOS open-access (CC-BY) license and provide specific permission for publication of these details under the terms of this license.

Please download the Consent Form for Publication in a PLOS Journal (http://journals.plos.org/plosone/s/file?id=8ce6/plos-consent-form-english.pdf). The signed consent form should not be submitted with the manuscript, but should be securely filed in the individual's case notes.

Please amend the methods section and ethics statement of the manuscript to explicitly state that the patient/participant has provided consent for publication: “The individual in this manuscript has given written informed consent (as outlined in PLOS consent form) to publish these case details”.

If you are unable to obtain consent from the subject of the photograph, you will need to remove the figures and any other textual identifying information or case descriptions for these individuals.

Reviewers' comments:

Reviewer's Responses to Questions

**Comments to the Author**

1. Is the manuscript technically sound, and do the data support the conclusions?

Reviewer #1: No

Reviewer #2: Partly

2. Has the statistical analysis been performed appropriately and rigorously? 

Reviewer #1: Yes

Reviewer #2: I Don't Know

3. Have the authors made all data underlying the findings in their manuscript fully available?

Reviewer #1: Yes

Reviewer #2: Yes

4. Is the manuscript presented in an intelligible fashion and written in standard English?

Reviewer #1: No

Reviewer #2: No

5. Review Comments to the Author

Reviewer #1: The manuscript presented a study aimed to evaluate the relationship between kinesiophobia, pain, proprioception and functional performance among subjects with chronic neck pain. The influence of kinesiophobia in the musculoskeletal rehabilitation is well understood in the previous studies, however, the authors justified that limited studies have been done among people with CNP. Generally, the paper was written with a good flow, A few comments for improvement is suggested prior to publication:

1. Revise the title, may need to change the term subjects. Also be consistent in the paper either to use subjects or participants.

2. In the introduction, it was not clearly explained which one comes first, pain or kinesiophobia or vice versa. As it is clearly stated in the previous literature pain is the cause of impairment. This needs to be clearly discussed.

3. In the Methodology; some outcome measures were not well described in terms of its procedure, so that it is replicable.

4. In the results: The authors have not considered any confounding factors such as duration of condition, nature of job, etc.

5. Others: i. Some typo errors were noted, need to check for English as well.

Reviewer #2: In the current manuscript, authors have investigated the “Kinesiophobia and its correlations with pain, proprioception, and functional performance among subjects with chronic neck pain”. It is a very interesting study, but I strongly encourage the authors to edit their paper with someone who has professional proficiency in English. In some parts, I had some difficulty to understand what the authors are trying to communicate.

Reviewer Comments:

Title: Change the “subjects” to “individuals”

Introduction and discussion need to be written again and consider adding more up to date references. You can use key words like “neck pain and upper limb proprioception” in your literature review researches.

Methods:

• What is “patient information sheet”? Is it study procedure or any type of questionnaires?

• My suggestion is to change the subjects to participants for the entire of the manuscript

• What was the age range? You just mentioned: “All the participants included in this study were over 18 years and ...” for example you should say (18-50 years)

• In addition, say how many male and female were participated in your study.

• Page 6: “1) Subjects diagnosed with CNP and pain subsequently felt between the first thoracic spinal process and the occiput's lower margin” how did you diagnosed that?

• Page 6: “2) neck pain intensity of 30 to 7o mm measured on a visual analog scale (VAS).” Fix the number 70 and clarify why you choose this intensity?

• Page6: “3) moderate neck disability score, between 15 to 30 measured on neck disability index” according to the NDI scoring your value should be between 15 - 24 to be consider a moderate neck pain. Check the following website (https://chiro.org/LINKS/OUTCOME/Painter_1.shtml)

• For sample size calculation what was your rational to use Tempa scale of kinesiophobia (TSK) score to estimate the study sample.

• Page 7: for the Anthropometric Data, refer to the table that has these information

• Page 7? Why you have the results for the Kinesiophobia Assessment but you did not included the results for Pain assessment

• Page 8: you should explain here what is your rational for choosing (50 % of their maximum ROM)

• Page 8: did you only analysis the absolute error? How about constant and variable errors?

• Functional Performance need to be more clarified. For how long they hold the dynamometer. What do you mean with neutral position? What was the position of the shoulder? What do you meand with dominant side> Right hand or left hand? Why you chose the highest value for analysis? Why you din not average the trials? How many trials they did? Why one minute rest? You should explain all the procedure and rational of your work in here.

• Change the title of table 1 to “Participant demographics, self-report measures, and study data”

• Page 10 “Kinesiophobia showed a positive mild to a moderate positive correlation with proprioception JPE errors in extension” there is two positives

• You should add more information on how to use the CROM

• Improve your JEP section

section

6. PLOS authors have the option to publish the peer review history of their article (what does this mean?). If published, this will include your full peer review and any attached files.

Reviewer #1: **Yes: **Associate Professor Dr Maria Justine

Reviewer #2: No

---

## [Author Response · Author response to Decision Letter 0]

22 Apr 2021

Thank you for your effort and time in reviewing our manuscript. The reviewing process has significantly improved the quality of this manuscript. I am submitting this "Response to reviewers" document summarizing the changes that we made in response to the critiques.

RESPONSE TO REVIEWR 1

1. 1. Revise the title, may need to change the term subjects. Also be consistent in the paper either to use subjects or participants. 

Answer• The term “subjects” was replaced with “individuals.” The changes are implemented throughout the manuscript.

2. 2. In the introduction, it was not clearly explained which one comes first, pain or kinesiophobia or vice versa. As it is clearly stated in the previous literature pain is the cause of impairment. This needs to be clearly discussed. 

Answer• I agree with your statement that pain comes first, and it is the cause of impairment. • The introduction is modified.

3. 3. In the Methodology; some outcome measures were not well described in terms of its procedure, so that it is replicable. 

Answer• The outcome measures are described in detail to replicate the study procedure.

4. 4. In the results: The authors have not considered any confounding factors such as duration of condition, nature of job, etc. A

h as duration of the condition, nature of the job, etc., were not considered, as these were not the objectives of this study, but as you say, these can be the confounding factors. 

• The same is mentioned as a part of the limitations of this study.

5. 5. Others: i. Some typo errors were noted, need to check for English as well. 

Answer• Professional English experts have edited the manuscript to fix the grammatical issues. 

RESPONSE TO REVIEWR 2

1. Reviewer #2: In the current manuscript, authors have investigated the “Kinesiophobia and its correlations with pain, proprioception, and functional performance among subjects with chronic neck pain”. It is a very interesting study, but I strongly encourage the authors to edit their paper with someone who has professional proficiency in English. In some parts, I had some difficulty to understand what the authors are trying to communicate. 

Answer• Professional English editing experts have edited the manuscript. 

2. Reviewer Comments:

Title: Change the “subjects” to “individuals”

Answer• The term “subjects” was replaced with “individuals.” The changes are implemented throughout the manuscript.

3. Introduction and discussion - consider adding more up to date references. You can use key words like “neck pain and upper limb proprioception” in your literature review researches. 

Answer• The introduction and discussion rewritten and added with more up-to-date references.

4. Methods:

• What is “patient information sheet”? Is it study procedure or any type of questionnaires? 

Answer• The information sheet included a concise summary of the research project and its objectives and a detailed description of the research process. It defined what participation entails in practice, how long it takes, where it occurs, and what it entails.

5. • My suggestion is to change the subjects to participants for the entire of the manuscript 

Answer• The term “subjects” was replaced with “individuals.” The changes are implemented throughout the manuscript.

6. • What was the age range? You just mentioned: “All the participants included in this study were over 18 years and ...” for example you should say (18-50 years) 

Answer• All the participants included in this study were over 18 years (age range 18 to 64 years). • Changes are incorporated in the manuscript. 

7. • In addition, say how many male and female were participated in your study. 

Answer• No. of male and females participated in this study are mentioned in the text and in table 1. 

8. • Page 6: “1) Subjects diagnosed with CNP and pain subsequently felt between the first thoracic spinal process and the occiput's lower margin” how did you diagnosed that? 

Answer• I mean to say that subjects are diagnosed with neck pain by the orthopaedician, neurosurgeon, or general physician and 

• Neck pain was defined as pain felt dorsally between the inferior margin of the occiput and T1.

• The confusing statement is modified. 

9. • Page 6: “2) neck pain intensity of 30 to 7o mm measured on a visual analog scale (VAS).” Fix the number 70 and clarify why you choose this intensity? 

Answer• We included neck pain intensity of 30 to 70 mm measured on a visual analog scale (VAS) as we want to make sure all the subjects had mild to moderate neck pain intensity levels. 

• Previous studies also have used similar inclusion criteria -Lauche, R., Langhorst, J., Dobos, G. J., & Cramer, H. (2013). Clinically meaningful differences in pain, disability and quality of life for chronic nonspecific neck pain–a reanalysis of 4 randomized controlled trials of cupping therapy. Complementary therapies in medicine, 21(4), 342-347.

• Number 70 is fixed.

• Changes are incorporated in the manuscript. 

10. • Page6: “3) moderate neck disability score, between 15 to 30 measured on neck disability index” according to the NDI scoring your value should be between 15 - 24 to be consider a moderate neck pain. Check the following website (https://chiro.org/LINKS/OUTCOME/Painter_1.shtml)

Answer • I agree with you that according to the NDI scoring, the value should be between 15 - 24 to be considered a moderate neck pain

• There is a Typo error. We included subjects with NDI score of 15 or above. 

• The changes are incorporated in the manuscript. 

11. • For sample size calculation what was your rational to use Tempa scale of kinesiophobia (TSK) score to estimate the study sample. 

Answer• We selected the most important parameter (i.e., TSK score) in the study and computed the sample size required for it. The rest of the parameters were not able to provide enough precision.

• We computed the sample size separately for each parameter in the study and used the greatest obtained sample size i.e., TSK score. 

12. • Page 7: for the Anthropometric Data, refer to the table that has these information 

Answer• The changes are incorporated. 

• referred to Table 1 for information

13. • Page 8: you should explain here what is your rational for choosing (50 % of their maximum ROM) 

Answer• The tests were conducted in midranges (50 % of their maximum ROM) and therefore not engaging the joint receptors as limit detectors, highlighting the emphasis of tests for muscle spindle afferent activity. 

• Reference: Hillier, Susan, Maarten Immink, and Dominic Thewlis. "Assessing proprioception: a systematic review of possibilities." Neurorehabilitation and neural repair 29.10 (2015): 933-949.

• The cervical JPE measurement protocol was adopted from Alahamrirt et al. study. This study also used 50% of the maximum range of motion as target head position. 

• (Alahmari, Khalid, et al. "Intra-and inter-rater reliability of neutral head position and target head position tests in patients with and without neck pain." Brazilian journal of physical therapy 21.4 (2017): 259-267.)

14. • Page 8: did you only analysis the absolute error? How about constant and variable errors? • In this study, only absolute errors were recorded; constant and variable errors were not measured.

• The same has been mentioned as one of the limitations of this study. 

15. • Functional Performance need to be more clarified. For how long they hold the dynamometer. What do you mean with neutral position? What was the position of the shoulder? What do you meand with dominant side> Right hand or left hand? How many trials they did? Why one minute rest? You should explain all the procedure and rational of your work in here. 

Answer• The handgrip strength evaluation procedure is rewritten. This test is performed with the subject sitting in the chair with the shoulder adducted and neutrally rotated, elbow flexed to 90 degrees, forearm and wrist maintained in the neutral position (neither flexed nor extended) gripping the handheld dynamometer.

• The handgrip strength measurements were performed on the dominant side (mostly right hand), three trials were conducted, and an average of three attempts was used for data analysis.

• Between each attempt, a one-minute rest period was allowed to minimize fatigue effects. Previous studies also considered one-minute rest between trials, which was enough to eliminate the effects of fatigue. (Bobos, Pavlos, et al. "Measurement properties of the handgrip strength assessment: a systematic review with meta-analysis." Archives of physical medicine and rehabilitation 101.3 (2020): 553-565.)

16. • Change the title of table 1 to “Participant demographics, self-report measures, and study data” 

Answer• Title of table 1 is changed to “Table 1. Participant demographics, self-report measures, and study data (𝑛 = 64).”

17. • Page 10 “Kinesiophobia showed a positive mild to a moderate positive correlation with proprioception JPE errors in extension” there is two positives 

Answer• English editing of the article is done and the sentence is modified. 

18. • You should add more information on how to use the CROM 

Answer• Added more information on how to use CROM.

---

## [Editor Report · Decision Letter 1]

11 May 2021

PONE-D-21-05740R1

Kinesiophobia and its correlations with pain, proprioception, and functional performance among subjects with chronic neck pain

PLOS ONE

Dear Dr. REDDY,

Thank you for submitting your manuscript to PLOS ONE. After careful consideration, we feel that it has merit but does not fully meet PLOS ONE’s publication criteria as it currently stands. Therefore, we invite you to submit a revised version of the manuscript that addresses the points raised during the review process.

You have mainly addressed the reviewers' concerns however However simply say that only absolute errors were recorded as a limitation is not acceptable. Constant and Variable errors can be calculated from the data that you have. There are many articles with the formulas. Here is one of many that explains how to calculate absolute, constant and variable error. https://motorcontrol.wordpress.com/2008/06/19/constant-error-variable-error-absolute-error-and-root-mean-square-error-labview/#:~:text=The%20formula%20for%20it%20is,alone%20without%20mentioning%20the%20direction.

Please include all error types in the methods, results and discussion of the next revision.

We look forward to receiving your revised manuscript.

Kind regards,

Bernadette Ann Murphy, PhD

Academic Editor

PLOS ONE
---

## [Author Response · Author response to Decision Letter 1]

18 Jun 2021

· Thank you for sharing the link which showed the formulas to calculate the constant and variable error.

· We are unfortunate that we could not calculate the Constant and variable errors. 

The formula to measure constant error is: Σ (xi-T)/N, and variable error is sq. root (Σ (xi-M)^2/N. In the formula "Xi" indicates the deviation from the target; it comes with a positive (overshooting) or negative sign (undershooting) that points out the error's direction. We did not measure or record this overshooting or undershooting. We measured the total error around the target (absolute error) only.

---

## [Editor Report · Decision Letter 2]

24 Jun 2021

Kinesiophobia and its correlations with pain, proprioception, and functional performance among individuals with chronic neck pain

PONE-D-21-05740R2

Dear Dr. REDDY,

We’re pleased to inform you that your manuscript has been judged scientifically suitable for publication and will be formally accepted for publication once it meets all outstanding technical requirements.

Kind regards,

Bernadette Ann Murphy, PhD

Academic Editor

PLOS ONE
---

## [Editor Report · Acceptance letter]

28 Jun 2021

PONE-D-21-05740R2 

Kinesiophobia and its correlations with pain, proprioception, and functional performance among individuals with chronic neck pain 

Dear Dr. Reddy:

I'm pleased to inform you that your manuscript has been deemed suitable for publication in PLOS ONE. Congratulations! Your manuscript is now with our production department. 

Kind regards, 

on behalf of

Dr. Bernadette Ann Murphy 

Academic Editor

PLOS ONE